# Tensile and Bending Strength Improvements in PEEK Parts Using Fused Deposition Modelling 3D Printing Considering Multi-Factor Coupling

**DOI:** 10.3390/polym12112497

**Published:** 2020-10-27

**Authors:** Yao Li, Yan Lou

**Affiliations:** Guangdong Provincial Key Laboratory of Micro/Nano Optomechatronics Engineering, College of Mechatronics and Control Engineering, Shenzhen University, Shenzhen 518060, China; ly0306m@163.com

**Keywords:** poly-ether-ether-ketone (PEEK), fused deposition modeling (FDM), multi-factor coupling, mechanical strength, grain size

## Abstract

Compared with laser-based 3D printing, fused deposition modelling (FDM) 3D printing technology is simple and safe to operate and has a low cost and high material utilization rate; thus, it is widely used. In order to promote the application of FDM 3D printing, poly-ether-ether-ketone (PEEK) was used as a printing material to explore the effect of multi-factor coupling such as different printing temperatures, printing directions, printing paths, and layer thicknesses on the tensile strength, bending strength, crystallinity, and grain size of FDM printed PEEK parts. The aim was to improve the mechanical properties of the 3D printed PEEK parts and achieve the same performance as the injection molded counterparts. The results show that when the thickness of the printed layer is 0.1 mm and the printing path is 180° horizontally at 525 °C, the tensile strength of the sample reaches 87.34 MPa, and the elongation reaches 38%, which basically exceeds the tensile properties of PEEK printed parts reported in previous studies and is consistent with the tensile properties of PEEK injection molded parts. When the thickness of the printed layer is 0.3 mm, the printing path is 45°, and with vertical printing direction at a printing temperature of 525 °C, the bending strength of the sample reaches 159.2 MPa, which exceeds the bending performance of injection molded parts by 20%. It was also found that the greater the tensile strength of the printed specimen, the more uniform the size of each grain, and the higher the crystallinity of the material. The highest crystallinity exceeded 30%, which reached the crystallinity of injection molded parts.

## 1. Introduction

Fused deposition modelling (FDM), as the most common technology in additive manufacturing, not only has a low cost but also has very high practical value in the field of 3D printing [1,2,3]. Through the “bottom-up” layer-by-layer stacking method of FDM, a complex three-dimensional structure can be fabricated; this is especially true for special engineering plastics and functional gradient devices that are difficult to process [4]. FDM is widely used in the medical [5], electronic device [6], aerospace [7] fields as well as other fields, which has prompted scholars to further explore the FDM process [8]. The “layered effect” is a typical feature of FDM. Liang et al. [9] found that the step height changed with the error between the layers during FDM molding and established a compensation model through a change in the extrusion filament diameter to improve the surface quality of the printed parts. Chacon et al. [10] suggested that the selection of different construction directions has a great influence on the fracture strength, rigidity, and deformation of the printed part.

Although increasing the filament feed amount during printing can shorten the printing time, the higher the feed amount, the lower the tensile and bending properties of the sample; among printing directions, it is considered that the mechanical properties of vertical printing are the worst, and the mechanical properties of the side vertical and horizontal directions are an improvement. Ismail et al. [11,12] suggested that the grating angle and part orientation have a great influence on the mechanical strength and surface quality of the formed test piece. Under the premise of considering the production cost, the functional relationship is used to obtain the best process parameters and guide practical applications.

The materials formed by FDM printing are mainly acrylonitrile butadiene styrene (ABS) [13,14], polylactic acid (PLA) [15,16], and polycarbonate (PC) [17], which have a relatively low cost and low melting point. The special engineering plastic poly-ether-ether-ketone (PEEK), studied in this paper, is an expensive thermosetting thermoplastic with a melting point of 343 °C, with good biocompatibility, heat resistance, and chemical resistance. Through injection molding and other molding processes, we can manufacture structural components that can not only replace human organs [18,19,20] but also replace expensive metal engine parts [21] and aircraft metal tail connecting brackets [22,23]. However, the injection molding process of the PEEK material requires a matching mold for each molded part. The price of the molds is relatively expensive, and the temperature change in the molds has a great influence on the molding quality. There are strict process requirements that are complicated for complex arrangements. For components, the disassembly of the mold is also a problem, but FDM additive manufacturing solves this problem. The disassembly of FDM parts involves mainly the removal of the support structure [24]. For example, hydrosol can be inserted into the side of the double nozzle as the supporting structure during printing. After the printing work is completed, the supporting structure of the molded part can be removed by a water- soluble method [25]. Compared to injection molding, this greatly reduces production costs. The application of PEEK in 3D printing has not been extensive. Deng [26] optimized the PEEK printing parameters through a customized FDM system to obtain process parameters from the macro that produce optimal mechanical properties, providing guidance for subsequent research. Kumar et al. [27] established three different printing models and showed that the influence of different grating angles (H-0, H-90, and V-90) on the thermal gradient changed and the mechanical properties during the printing process increased in sequence. Li et al. [28] found that different printing directions and other multiscale factors affected the bending strength and crystallinity of printed PEEK.

However, the mechanical strength of 3D printed PEEK parts is far lower now than that of the injection parts. Pure PEEK injection molded parts can reach a tensile strength from 80–100 MPa [29,30,31], and the PEEK crystallinity can reach 30% [32]. However, the tensile strength of 3D printed PEEK parts is generally only 50 MPa [33,34,35,36], which is far lower than that of the injection parts. Therefore, this study aimed to continuously improve the mechanical properties of FDM printed PEEK parts by studying the effects of layer thickness, temperature, printing path, and printing direction on the tensile strength, bending strength, crystallinity, and grain size of PEEK parts. Differential scanning calorimetry (DSC) and X-ray diffraction (XRD) were used to further explore the properties of the FDM printed PEEK parts and provide reference information for obtaining the best mechanical properties for the FDM printed PEEK parts.

## 2. Experimental Methods

### 2.1. Equipment and Materials

The equipment used in this experiment comprised an Apium P220 FFF 3D printer with an adaptive heating system developed by the Apium Company (Karlsruhe, Germany). The X/Y resolution was 0.0125 mm, and the Z resolution 0.05 mm. The specific printing process parameters in this study are shown in Table 1. As the temperature of the printing bed can reach 160 °C, in addition to setting the three layers of Dimafix glue, the hemlines should be set before printing to prevent the bottom edge of the printed piece from sticking out. The key step before the start of the experiment is to level the gap between the nozzle and the print bed multiple times until a piece of standard A4 paper can be moved in the gap; the experiment can begin when there is a little resistance. The filament for the experiment is an Apium PEEK 450 natural filament produced by Apium with a diameter Φ of 1.75 ± 0.05 mm. The glass transition of this PEEK filament is 143 °C. To prevent bubbles from impacting the prints and the molding, the PEEK filament needs to be dried at 120 °C for 4 h before the experiment [37]. After the experiment, the part was cooled at room temperature without subsequent processing.

During the experiment, SolidWorks (2018 edition) was used first to model the printed sample, then the STL format file of the printed sample was imported to allow the 3D slicing software to set the print parameters. Finally, the converted G-CODE was entered into the Apium P220 printer so that it could read the print instructions. To focus on the influence of printing layer thickness, nozzle temperature, printing direction, and printing path on the mechanical properties of FDM printed PEEK parts, the test was carried out at a filling density of 100% to prevent the formation of pores at the interlayer between filaments [38,39]. PEEK samples were printed at a speed of 30 mm/s, a substrate temperature of 115 °C, and a nozzle diameter of 0.4 mm. According to a performance test of Dimafix glue, the optimal viscosity temperature is approximately 115 °C. Therefore, the temperature of the print bed was set to 115 °C. By changing the layer thickness, nozzle temperature, printing direction and printing path, the mechanical properties of FDM printed PEEK parts were studied (Figure 1). The printing direction of the printed PEEK parts can be divided into the horizontal and vertical direction. The printing path is set as the angle between the nozzle moving direction and the Y-axis. To achieve better regularity study, 45°, 90°, and 180° were selected as the printing path. They represent the angles between the nozzle moving direction and the Y-axis which are 45°, 90°, and 180°, respectively. Although the melting temperature of the PEEK material is 343 °C, the high viscosity and poor fluidity at this temperature result in the PEEK not being printed. In order to make the filaments normally extrude without blocking the nozzle and to provide a good interlayer adhesion effect, a printing temperature of 445 °C or higher was selected to investigate the performance of the molded parts. Orthogonal experimental schemes were selected, as shown in Table 1.

After the orthogonal design, the mixed orthogonal test group L_9_ (2 × 3^3^) (Table 2) has a total of nine groups of printing parameters. Each group of parameters prints five repeat samples. The mechanical properties and crystallinity of PEEK parts under different printing conditions were studied by collecting the data and equalization processes.

### 2.2. Tensile Test

An automatic electronic universal testing machine (Z050T, Zwick/Roell, Berlin, Germany) was used for the tensile testing of standard tensile test samples (ISO527-2:1993 1BA). The test speed was 1 mm/min, and the maximum loading force was 50 kN. The temperature was 25 °C in a room-temperature environment. To improve the reliability of the results, five samples were selected for each group of parameters in this study for statistical analysis.

### 2.3. Bending Test

To fully understand the mechanical properties of PEEK, a bending test was required. The equipment used was an electronic universal testing machine (SANS, CMT5504, MTS Industrial System, Shenzhen, China). The test was performed at a pressing speed of 1 mm/min. The bending test of a standard bending test sample (ISO178:2001) was carried out. The maximum loading force was 50 kN, and the span of the bending test was 30 mm.

### 2.4. Differential Scanning Calorimetry (DSC)

The heat resistance and thermal stability of PEEK specimens can be judged by the crystallinity and grain size, and the relationship between the crystallinity and the mechanical properties of the specimens can be explored. The heat of enthalpy can be obtained by differential scanning calorimetry (DSC, DSC8000, Perkin Elmer Instruments, Norwalk, CT, USA), and the area surrounded by the melting peak curve and the baseline can be directly converted into heat, which is the heat of fusion of the polymer material. The heat of fusion of a polymer is directly proportional to its crystallinity, so the heat of fusion of the polymer was calculated to determine the crystallinity. The specific calculation formula of the crystallinity is as follows:(1)$$\theta =\frac{∆H_{f}}{∆H_{f}^{*}}\times 100\%$$
where *θ* is the crystallinity (unit: %); $∆H_{f}$ is the heat of fusion of a pure PEEK polymer material, which varies and is a function of the processing conditions; $∆H_{f}^{*}$ is the heat of fusion when the pure PEEK crystallizes and is 130 J/g, according to the literature [40].

In this experiment, an electronic balance (HZY-A200, Huazhi, Hartford, CT, USA) was used to weigh the sample. The selected 6 g sample was put into the differential scanning machine and heated at 20 °C/min, and the enthalpy change of the sample was observed when it was heated from 100 °C to 500 °C. In the same sample, the crystallinity of the core area and the surface area were different [28]. Therefore, to avoid environmental factors affecting the data change of the specimen, the DSC experiment sample was selected from the core area.

### 2.5. X-ray Diffraction (XRD) and Scanning Electron Microscopy (SEM)

X-ray diffraction (MiniFlex600, Rigaku, Tokyo, Japan) was used for characterization and performance analysis. The pure PEEK is composed of mainly C, H, and O. Because the grain size was large and the order was poor, there were only X-ray diffraction peaks at low angles. Thus, the experimentally set diffraction angle (2θ) ranged from 2–50 °C, the scanning speed was 5°/min, and the tested sample was the tensile fracture surface of the tensile specimen. The anode was Cu, and the software for diffraction data processing was MDI jade6. With this software, the size and crystallinity of the grains of the print can be calculated. Thermal field emission scanning electron microscopy (SU-70, Hitachi, Tokyo, Japan) was used to characterize and observe the fractures of the tensile sample. The grain size of the sample was quantified under different parameters.

## 3. Results and Discussion

### 3.1. Tensile Strength

Table 3 shows the range analysis of orthogonal design. During the tensile test, factor A (layer thickness) had the largest range; that is, the layer thickness had the greatest influence on the tensile strength, and a layer thickness taken as 1 level (0.1 mm) was better. The second point is the influence of factor C (printing direction) on the tensile strength, and parallel printing was better. The range of factor B (temperature) was the smallest; that is, the temperature had the least influence on the tensile strength.

It can be seen in Figure 2 that the tensile properties of sample a are the best when the layer thickness–temperature–printing direction–printing angle parameters are 0.1–525 °C–H–180° (curve of b in Figure 2). The tensile strength reaches 87.34 MPa. It is worth noting that the printed sample under this condition has good plasticity and reaches a strain of 37.8%. The strength of PEEK parts in this paper reached 87.34 MPa, which is about 15% higher than reported in previous studies [41,42]. The reason is likely that the printing temperature of FDM in this study was higher, the filaments could be fully melted and bonded firmly, and achieved a higher strength.

The reason is that when the temperature is high enough and the layer thickness is small, the time for the entire cross-section of the filaments to pass through the heat transfer is short, so that the filaments ejected from the nozzle can have a better temperature homogenization and achieve a stable molten state. The horizontal printing method allows the first layer of the filaments printed on the substrate to produce a good setting effect and surface quality with sufficient cooling time, without considering the influence of other factors. Each filament does not break when printing in the same direction, and the thermal stress and temperature distribution of each segment of the filament for each direction in a single direction are relatively uniform.

As shown in Figure 3a, the temperature of X node fluctuates nearly periodically over time. This is because when the next filament is printed in the reverse direction on the same layer, the previous filament is thermally diffused because of the heat source of the nozzle, so that the instant the nozzle moves over, the previous filament that has not been completely cooled receives the heat source energy and is in a semi-melted state that can integrate with the next filament, and so on. The printing of each filament actually optimizes the filament and the previous filament. The degree of fusion between the filaments increases the adhesion between the filaments. In the same way, when multiple filaments are fused into one layer, the next layer begins. At this moment, as the heat source, the nozzle moves past each point and not only affects the semi-melted state of the previous filament but also affects the semi-molten state of the filament that was deposited in the previous layer. The superposition of multiple layers increases the mechanical properties and surface quality.

When exploring the tensile properties, the direction of the loading force in the tensile experiment is parallel to the printing direction of the filaments (sample a). When stretching, countless filaments elongate. When the printing temperature is 525 °C, the filaments are fused together well, resulting in good plasticity when sample a is stretched.

In addition, sample b with a tensile strength of 81.9 MPa (curve b in Figure 2) printed at 0.1–445 °C–H–45° has a better tensile strength. Compared with those for sample a, its temperature is reduced by 80 °C, and the printing path is 45°. The tensile strength is only reduced by approximately 6%, indicating that the temperature and printing path have an effect on the tensile strength of the sample, but the effect is not large.

Second, the printed sample with this parameter set has a very low plasticity, and there are probably two reasons. One is that the printing temperature of sample b is 445 °C, and that the filament in the nozzle has not been fully heated and converted into a uniform melted state when it is extruded out of the nozzle. The temperature inside the filament is still in the pre-melted state. The outer surface of the filament solidifies quickly when it encounters a baseplate whose temperature is much lower than that of the filament. Uneven temperature inside and outside causes poor plasticity; another reason is the printing path of sample b, as shown in the SEM image of the fracture of the tensile part (Figure 4).

Comparing the cross-section of sample a under the 180° printing path (Figure 4a), when the printing path is 45° (Figure 4b), there is a 45° angle between the tension direction of the sample and the travel path of the filament during the tensile test. Thus, the sample has to bear the component of the adhesion force between the filaments in the parallel direction of the tensile force and the connection of the filament itself. As a result of the two force dispersion effects, the mechanical properties of sample b under the 45° printing path are weaker than sample a under the 180° printing path.

The poor tensile strength is obtained for the 0.2–445 °C–V–180° parameter set (sample c in Figure 2). Its strength is only 43.8 MPa. Compared with sample a and sample b, it is obvious that the printing direction is vertical. In the tensile test, when the vertically printed sample is subjected to the tensile force, a large part is derived from the mutual adhesion between the layers. As shown in Figure 3b, there is mutual adhesion or insufficient adhesion between the layers and the filaments. In the case of insufficient adhesion, it produces voids. The loading of tensile force will cause dislocation and movement between layers, resulting in the final broken cross-section that is not perpendicular to the applied load direction but is parallel to the direction of the applied load. The debonding between the layers appears (see Figure 4c).

On the other hand, the tensile strength of a single PEEK filament can range from 90–100 MPa; after the additive manufacturing process, the tensile strength is slightly reduced due to the influence of thermal stress, but it is not only 40 MPa. The reason is that when the printing temperature is 445 °C, the filament cannot be completely melted, which leads to insufficient adhesion between filaments. This makes the interlayer adhesion weak, resulting in interlayer dislocation debonding, and the tensile strength is low.

Finally, sample d (0.3–525 °C–V–45°) has the worst tensile strength, which is due to the layer thickness being large and the printing direction vertical. Although there is full melting between the layers at high temperature, the cross-section of the tensile fracture is relatively flat, and there is obvious brittle fracture. This is consistent with the analysis results in Table 4. The layer thickness has the greatest influence on the tensile properties of the sample, and the printing direction is second.

### 3.2. Bending Strength

Table 4 shows the range analysis of orthogonal experimental design for bending strength. In the bending test, the factor C (printing direction) has the greatest influence on the bending strength, and level 2 (vertical printing direction) is better. The influence of factor D (printing path) is second, and the influence of factor A (layer thickness) is the smallest, which is completely different from the influence of the tensile strength.

It can be seen in Figure 5 that sample d according to the parameter group 0.3–525 °C–V–45° has a bending strength of 159.2 MPa when the bending strain is 0.08%, which is the largest bending strength among the parameter groups. Li et al. [28,43] studied the bending performance of PEEK molded parts under different process conditions through process optimization, and found that the bending strength of pure PEEK injection molded parts was 130 MPa, and the bending strength of 3D printed parts was 110–150 MPa. The bending strengths of 3D printed parts studied here have been improved compared to those of their predecessors [28].

The parameter set is 0.1–485 °C–V–90° (sample e in Figure 5), where the bending strength reaches 153.6 MPa. The main reason for this is that the surface area of a single layer printed in the vertical direction is smaller than that in the horizontal printing. Therefore, the time required to print a single layer vertically is shorter than the time required to print a single layer horizontally, resulting in a smaller thermal gradient effect between layers of vertical prints compared to between layers of horizontal prints. A smaller thermal gradient makes the interlayer adhesion greater, so the bending strength of vertical prints under the same process parameters is generally higher than that of horizontal prints. This is consistent with the views of Li et al. [28]. Second, the printing temperature is high. The minimum printing temperature of sample d and sample e is 485 °C, which is enough to make the interlayer adhesion better.

In addition, it is worth noting the stress–strain curve of sample c (0.2–445 °C–V–180°) in Figure 5. When the strain is greater than 0.2%, the instantaneous decrease in the stress value occurs continuously. According to the stress shielding effect, under bending load, the neutral surface of the sample moves downwards with the increase of flexion, so that the force on the bottom of the bending piece is relatively reduced, resulting in in-plane bending. It is easy to damage the adhesive properties, which l causes interlayer dislocation and debonding under the action of tension on the bottom surface and pressure on the top surface. This will reduce the bending strength.

Furthermore, the plasticity of sample a (0.1–525 °C–H–180°) and sample c are very good, but the fracture mode of sample a is ductile fracture, which is completely different from the interlayer debonding mode of sample c. Because the printing layer thickness of sample a is small and the printing temperature is high, the material is readily in a completely molten state, which makes the adhesive effect between the layers very good. The neutral surface of sample a is subject to bending force, but due to the good adhesion of the materials, the bottom of the sample is pulled and the top pressed to the neutral surface and drops to the critical point, and there is no interlayer debonding fracture.

The worst bending strength is sample f (0.3–445 °C–H–90°). When the strain is 0.09%, the bending strength is only 47.8 MPa. The printing direction of sample f, sample a, and sample b are all horizontally printed, but the bending strength of sample f is very low. One reason is that the temperature of sample f is 445 °C and the printing layer is 0.3 mm. The filament cannot be completely melted because of the low temperature and layer thickness, resulting in poor adhesion between the layers. On the other hand, when the printing path is 90°, the bending direction is exactly perpendicular to the filament, and the bending direction with a printing path of 180° is parallel to the filament, so sample f has the lowest strength with brittle fracture, while sample a has high bending strength and good plasticity, followed by sample b.

### 3.3. Crystallinity

When the temperature reaches a certain value, the PEEK material undergoes thermal decomposition, with the main mode being degradation. When the temperature exceeds 450 °C in the DSC experiment, the enthalpy curve has an upward trend, which indicates normal thermal decomposition of the material. The DSC thermal analysis images taken at temperatures from 100–450 °C can be seen, and the crystallinity of the sample after FDM printing is higher than that of the original material. Under the same printing layer thickness, as the printing temperature increases, the crystallinity of the sample increases, but the change range is relatively small; that is to say, different printing temperatures do not substantially affect the crystallinity of the FDM printed sample. Generally, if the crystallinity increases, the heat resistance and thermal stability increase, the molecular chain spacing of the crystal grains becomes regular, and the strength increases.

As shown in Figure 6, the crystallinity of sample a (0.1–525 °C–H–180°) is higher than that of sample b (0.1–445 °C–H–45°), and the tensile strength and plasticity of sample a are all better than sample b (Figure 2). This indicates that the crystallinity of the material is positively related to the tensile strength of the sample and if the printing direction is the same, the sample with higher crystallinity has better plasticity. The reason for this is that when the printing path is 180°, the force direction of the sample is the same as the horizontal in the direction of each filament, and the filament is fully melted at high temperature, making the crystal more uniform resulting in the tensile strength and plasticity being better. Similarly, after comparing the performance of sample c (0.2–445 °C–V–180°) and sample d (0.3–525 °C–V–45°), it was found that the sample with higher crystallinity had higher tensile strength and plasticity.

### 3.4. Crystallinity

The X-ray diffraction data was obtained, and the pure PEEK material mainly showed six typical crystal peaks [44]. Under different printing parameters, the diffraction intensity of the peaks was different, but the peak intensity from the (110) planes was the maximum and that from the (223) planes the minimum. On this basis, quantitative analysis of the grain size of the PEEK printed parts was conducted with jade software, and the crystallinity of the PEEK printed parts was 20–40%, which is consistent with the DSC experimental results.

The relationship between the grain size of the molded part and its mechanical properties was also studied. The grain size was determined from the XRD data processed by the jade software as follows:(2)$$L_{hkl}=\frac{K\lambda }{FW\left(S\right)*\cos\left(\theta \right)}$$
where $L_{hkl}$ represents the grain size (nm); *K* is the Scherrer shape factor and generally *K* = 1; λ is the X-ray wavelength (nm); *FW*(*S*) is the half-height width of the sample, and generally 0.89; and θ is the diffraction angle (Rad).

As shown in Figure 7b, the grain size in the samples under different printing parameters differs by 10–50 nm. To reflect the uniformity of the distribution of the grain size, the difference between the maximum value change of the grain size was determined and the analysis of the size variance conducted for each set of printing parameters. The smaller the difference, the smaller the size change, and the smaller the change, the higher the uniformity of the grain size, all which improve the mechanical properties of the sample.

The difference in the maximum grain size of sample a with the 0.1–525 °C–H–180° parameter set was determined to be only 2.6 nm, and the variance 0.9295. This indicates that among the selected parameter sets, sample a has the smallest change in the grain size, a more uniform grain size distribution, and better mechanical properties. For sample d with the worst tensile strength produced with the 0.3–525 °C–V–45° parameter set, the maximum difference in grain size reaches 5.9 nm, and the variance also reaches 4.279. This is a large variation in the grain size, and the sample had the worst mechanical strength among the samples herein, which was consistent with the tensile property results in Figure 2.

## 4. Conclusions

Considering multi-factor coupling used for fused deposition modelling (FDM) 3D printed PEEK molded parts, the printing parameters for the best mechanical properties were explored. On this basis, the crystallinity of the printed sample and the relationship between the size of the crystal grains and the mechanical properties of the sample were studied. The data show that the tensile strength of FDM printed PEEK parts can reach 87.6 MPa, the bending strength can reach 159.2 MPa, and the crystallinity can reach 32.4%. Basically, the mechanical properties can match those of injection molded parts, which not only enriches the forming process of PEEK materials but also provides the feasibility for further implementation of additive manufacturing instead of subtractive manufacturing.

Different parameter groups have a greater influence on the performance of the printed parts, especially for pure PEEK materials with strong inertia. At present, it is difficult to obtain the optimal processing parameters to solve all the problems, for examples, problems of edge warping and ladder effect of 3D printed parts have not been satisfactorioy solved.

## Figures and Tables

**Figure 1 polymers-12-02497-f001:**
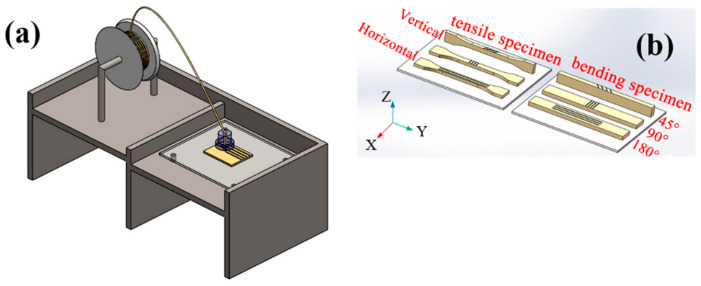
Schematic diagram of the FDM (Fused Deposition Modelling) experiment plan: (**a**) shows the principle diagram of the FDM printing device and (**b**) shows the different printing direction and path of the tensile and bending specimen: horizontal and vertical represent the printing direction; 45°, 90°, and 180° represent the printing paths, which means the angles between the nozzle moving direction and the Y-axis are 45°, 90°, and 180°, respectively.

**Figure 2 polymers-12-02497-f002:**
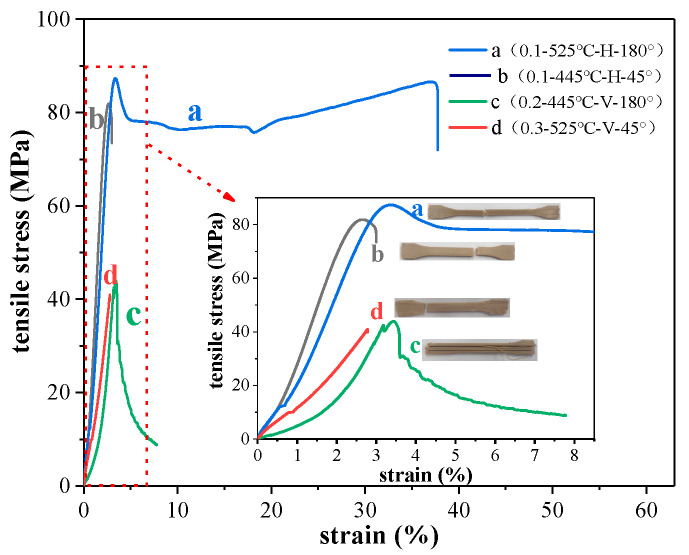
Stress–strain diagram and tensile fracture diagram of tensile specimen (where a represents 0.1–525 °C–H–180°; b represents 0.1–445 °C–H–45°; c represents 0.2–445 °C–V–180°; d represents 0.3–525 °C–V–45°).

**Figure 3 polymers-12-02497-f003:**
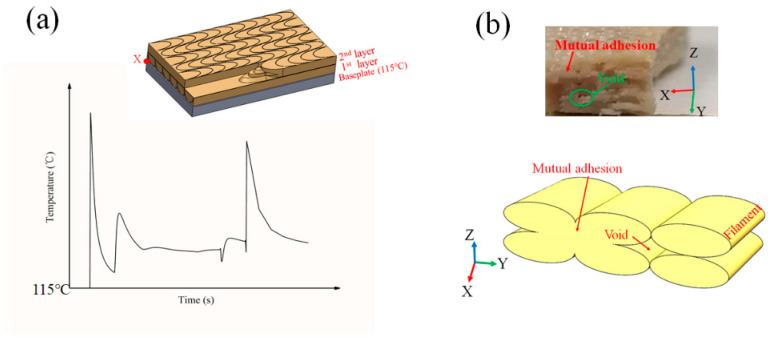
(**a**) The temperature change law at X node of FDM prints with time; (**b**) schematic diagram of the porosity of the fracture cross section.

**Figure 4 polymers-12-02497-f004:**
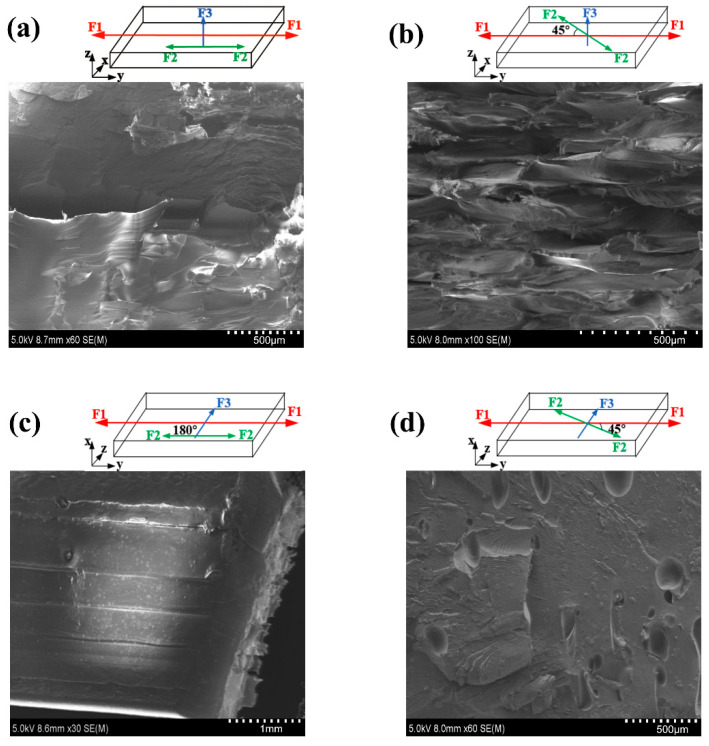
The fracture mode of the tensile specimen and the corresponding SEM tensile fracture morphology (**a**) 0.1–525 °C–H–180° ductile fracture; (**b**) 0.1–445 °C–H–45° semi-brittle fracture; (**c**) 0.2–445 °C–V–180° interlayer debonding (**d**) 0.3–525 °C–V–45° brittle fracture. (F1 represents the direction of tensile force; F2 represents the printing path; F3 represents the printing direction).

**Figure 5 polymers-12-02497-f005:**
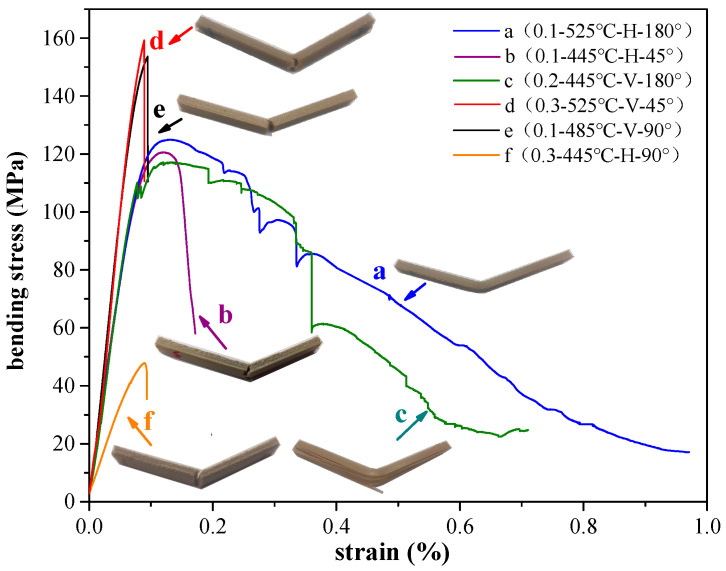
Stress-strain diagram and bending fracture diagram of the bending specimen (where a represents 0.1–525 °C–H–180°, ductile fracture; b represents 0.1–445 °C–H–45°, semi-brittle fracture; c represents 0.2–445 °C–V–180°, interlayer debonding; d means 0.3–525 °C–V–45°, brittle fracture; e represents 0.1–485 °C–V–90°, brittle fracture; f means 0.3–445 °C–H–90°, brittle fracture).

**Figure 6 polymers-12-02497-f006:**
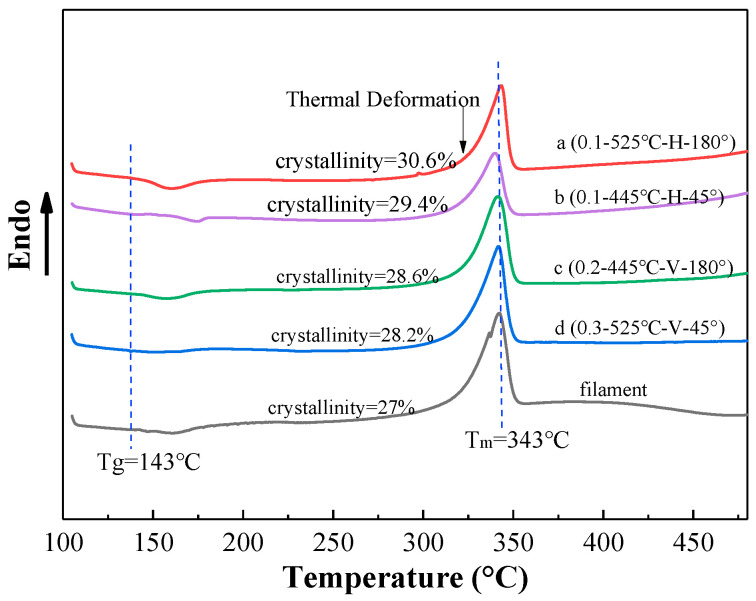
DSC heat flow curve.

**Figure 7 polymers-12-02497-f007:**
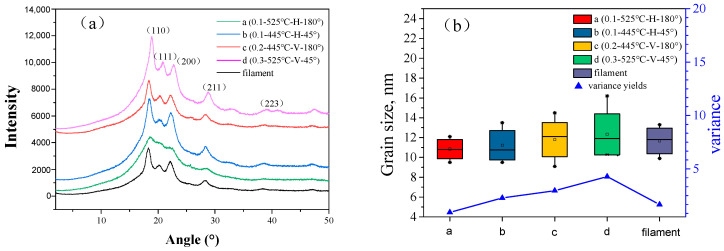
(**a**) XRD data graph; (**b**) grain size variance and maximum difference.

**Table 1 polymers-12-02497-t001:** Orthogonal experimental design of 3D printing.

Level	Factor			
	Layer Thickness (mm) A	Printing Temperature (°C) B	Printing Direction C	Printing Path (°) D
1	0.1	445	horizontal	45
2	0.2	485	vertical	90
3	0.3	525	-	180

**Table 2 polymers-12-02497-t002:** L_9_ (2 × 3^3^) hybrid quasi-horizontal orthogonal design.

No.	Factor			
	A	B	C	D
1	0.1	445	H	45
2	0.1	485	V	90
3	0.1	525	H	180
4	0.2	445	V	180
5	0.2	485	H	45
6	0.2	525	H	90
7	0.3	445	H	90
8	0.3	485	H	180
9	0.3	525	V	45

Note: H stands for horizontal printing direction, V stands for vertical printing direction.

**Table 3 polymers-12-02497-t003:** Range analysis of tensile strength.

Range	A	B	C	D
K_1j_	224.3	171.7	398.6	203.1
K_2j_	169.9	186	147.1	153.9
K_3j_	151.5	188	-	188.7
k_1j_	74.77	57.23	66.43	67.7
k_2j_	56.63	62	49.03	51.3
k_3j_	50.5	62.67	-	62.9
R	24.27	5.44	17.4	16.4
Order of priority	A C D B			

Note: K represents the total tensile strength under the same level i (i = 1, 2, 3), $k_{i}=\frac{K_{i}}{level\;number}$, R is range value: R_j_ = max{k_1j_, k_2j_, k_3j_} − min{k_1j_, k_2j_, k_3j_}, j is the variable factor, which stands for factor A, B, C, D.

**Table 4 polymers-12-02497-t004:** Range analysis of bending strength.

Range	A	B	C	D
K_1j_	339	285.6	589.5	390
K_2j_	309.6	373.5	423.9	277.8
K_3j_	304.8	354.3	-	345.6
k_1j_	113	95.2	98.25	130
k_2j_	103.2	124.5	141.3	92.6
k_3j_	101.6	118.1	-	115.2
R	11.4	29.3	43.05	37.4
Order of priority	C D B A			

Note: K represents the total bending strength under the same level i (i = 1, 2, 3), $k_{i}=\frac{K_{i}}{level\;number}$, R is range value: R_j_ = max{k_1j_, k_2j_, k_3j_} − min{k_1j_, k_2j_, k_3j_}, j is the variable factor, which stands for factor A, B, C, D.

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
