# Peer review of "Tensile and Bending Strength Improvements in PEEK Parts Using Fused Deposition Modelling 3D Printing Considering Multi-Factor Coupling"

_polymers, 2020, doi:10.3390/polym12112497_

Round 1
Reviewer 1 Report
The manuscript title “Tensile and Bending Strength Improvements in PEEK Parts Using FDM 3D Printing considering multi-factor coupling” submitted to “polymers” for publication was reviewed. In this paper, authors have investigated the effects of layer thickness, temperature, printing path and printing direction on the tensile strength, bending strength, crystallinity and grain size of PEEK using various characterization techniques including differential scanning calorimetry and X-ray diffraction. The manuscript fits well within the scope of the journal, however needs some improvements; there are a few suggestions that authors may consider to improve it further:
Abstract: is unstructured and very much descriptive; authors should clearly add the objective of the study and conclusive remarks.
Using FDM in the title is not clear; instead “Fused deposition modelling” can be used in the title.
Line 79 should read as “Equipment and materials”
Line 90: did authors followed any protocol for this statement? Please cite a reference
Table 1: is not needed really, the information present in table one can be stated as a text line.
Figure 1b: what is indicated by angles? Please add details to the captain,
Line 111: 3D
Line 121 and 131: “was used” authors are suggested to used correct form of English throughout the manuscript. There are several such mistakes in the methods section.
There is no mention of data handling and statistical analysis; could authors include such details in the methods section
Figure 4: Please improve scale bars using a bold scale; as the current scale bars are not clear
Line 258: why is it written in capital letters? Please correct.
Line 164: please change to results and discussion.
The section 3 is mainly results description and seems the discussion is a bit deficit; only a limited studies been included in the context. Authors should compare the findings with the previous similar studies in context. Is there any limitation of the study? There is hardly any discussion about the limitations of the study
Author Response
Dear reviewers,
We deeply appreciate the time and effort you’ve spent in reviewing our manuscript, “Tensile and Bending Strength Improvements in PEEK Parts Using Fused Deposition Modelling 3D Printing considering multi-factor coupling”. The comments of the reviewers really aid us to improve the paper. The main changes in our new manuscripts are highlighted in yellow color in the revised manuscript and we have provided a point-by-point response to reviewer’s comments. Please see the attachment.

Reviewer 2 Report
I have attached the report

Author Response

(The authors gave the same response as above.)

Round 2
Reviewer 1 Report
Dear Authors,
Many thanks for the revision and incorporating all suggested changes to the manuscript”